# Untargeted Metabolomics Identifies a Novel Panel of Markers for Autologous Blood Transfusion

**DOI:** 10.3390/metabo12050425

**Published:** 2022-05-10

**Authors:** Amna Al-Nesf, Nada Mohamed-Ali, Vanessa Acquaah, Maneera Al-Jaber, Maryam Al-Nesf, Mohamed A. Yassin, Nelson N Orie, Sven Christian Voss, Costas Georgakopoulos, Rikesh Bhatt, Alka Beotra, Vidya Mohamed-Ali, Mohammed Al-Maadheed

**Affiliations:** 1Anti-Doping Lab Qatar, Doha P.O. Box 27775, Qatar; aalnesef@adlqatar.qa (A.A.-N.); maaljaber@adlqatar.qa (M.A.-J.); n.orie@ucl.ac.uk (N.N.O.); svoss@adlqatar.qa (S.C.V.); costas@adlqatar.qa (C.G.); abeotra@adlqatar.qa (A.B.); vali@adlqatar.qa (V.M.-A.); 2Centre of Metabolism and Inflammation, UCL, London NW3 2PF, UK; nada.mohamed-ali.20@ucl.ac.uk (N.M.-A.); vanessa.acquaah.16@ucl.ac.uk (V.A.); 3Hamad Medical Corporation, Doha P.O. Box 3050, Qatar; mariamali@hamad.qa (M.A.-N.); yassinmoha@gmail.com (M.A.Y.); 4Department of Applied Health Research, UCL, London WC1E 7HB, UK; rikesh.bhatt@ucl.ac.uk

**Keywords:** untargeted metabolomics, autologous transfusion, metabolites, biomarker

## Abstract

Untargeted metabolomics was used to analyze serum and urine samples for biomarkers of autologous blood transfusion (ABT). Red blood cell concentrates from donated blood were stored for 35–36 days prior to reinfusion into the donors. Participants were sampled at different time points post-donation and up to 7 days post-transfusion. Metabolomic profiling was performed using ACQUITY ultra performance liquid chromatography (UPLC), Q-Exactive high resolution/accurate mass spectrometer interfaced with a heated electrospray ionization (HESI-II) source and Orbitrap mass analyzer operated at 35,000 mass resolution. The markers of ABT were determined by principal component analysis and metabolites that had *p* < 0.05 and met ≥ 2-fold change from baseline were selected. A total of 11 serum and eight urinary metabolites, including two urinary plasticizer metabolites, were altered during the study. By the seventh day post-transfusion, the plasticizers had returned to baseline, while changes in nine other metabolites (seven serum and two urinary) remained. Five of these metabolites (serum inosine, guanosine and sphinganine and urinary isocitrate and erythronate) were upregulated, while serum glycourdeoxycholate, S-allylcysteine, 17-alphahydroxypregnenalone 3 and Glutamine conjugate of C_6_H_10_O_2_ (2)* were downregulated. This is the first study to identify a panel of metabolites, from serum and urine, as markers of ABT. Once independently validated, it could be universally adopted to detect ABT.

## 1. Introduction

The use of blood transfusions by elite athletes, as a means of enhancing sports performance has been well documented [1,2]. Early reviews associating blood transfusions with improved athletic performance were conflicting, with some claiming that the increases in blood volume and oxygen delivery post-transfusion were too transient to be advantageous [3,4]. There is now growing evidence that blood transfusions lead to enhanced aerobic power and endurance performance, primarily through increased oxygen-carrying capacity due to enriched haemoglobin (Hb) concentration [3,5]. In addition, the elevated red blood cell (RBC) mass increases lactic acid buffering and thermoregulation [3,5,6,7]. As a result, blood transfusions, both autologous and homologous, are prohibited by the World Anti-Doping Agency (WADA) [8].

Homologous blood transfusions (HBT) refers to the process in which blood is taken from a donor and transfused into a different recipient. HBTs can be detected, either by identifying differences in expression patterns of minor blood group antigens [9,10], or via analysis of DNA short tandem repeats (STRs) [11,12]. Autologous blood transfusion (ABT) is the process of collecting, storing and re-infusing whole blood or RBC concentrates into the same donor individual. It is a common surgical practice, as it alleviates the risks associated with allogenic blood exposure, such as the spreading of infectious diseases and immune reactions [13,14]. However, in the context of sports doping, detection of ABTs has proven challenging. Thus, identifying robust, reliable methods of detection is an area of intense research activity.

Currently, ABT can only be detected indirectly, via changes in the Athletes Biological Passport (ABP). The ABP is a longitudinal blood profile, recorded in a central database over time for each individual. It is based on haematological parameters such as haemoglobin concentration and reticulocyte percentage, combined with heterogenous factors such as age, ethnicity and sex, and allows for the detection of variations in these parameters outside of the physiological norms [15]. While useful as a tool, the ABP is not without limitations, and is confounded by variables such as plasma volume variations, hyperhydration and altitude/hypoxic training [16,17,18,19].

In recent years, there has been a shift away from indirect methods of detection, towards uncovering biomarkers for ABT. Candidate biomarkers include RBC microparticles that have been shown to increase 100-fold following storage of whole blood for 14 days. While this is an interesting biomarker for blood storage, it is less useful post-transfusion [20]. Highly significant increases in RBC-derived extracellular vesicles have been reported post-transfusion, albeit with individual variability and a detection limit of 8 h [21]. In T-lymphocytes, ABTs can alter the expression of target genes for up to 96 h after transfusion. Nevertheless, this method lacks the capacity to differentiate between ABTs and immune reactions to infections or haemolysis [22]. Analysis of circulating miRNAs have been investigated, but did not show superior diagnostic potential compared to ABP. Perhaps a more extensive miRNA pool of analysis may be required [23,24].

Recent investigations have proposed “omics” based strategies to uncover promising, new biomarkers for ABTs. The expression of three reticulocyte genes relating to erythropoiesis have been identified following ABT via transcriptomics [25]. Using proteomics, changes in RBC membrane proteins have been observed following storage of whole blood, although it’s unclear how long this perturbance is detectable [26,27]. While encouraging, these approaches still face challenges in terms of identifying biomarkers with longer detection windows.

Of late, there has been increasing interest in using untargeted metabolomics for detection of ABT. In blood, vasoactive, pro-oxidative, proinflammatory and immunomodulatory metabolites were detected after 42 days of cold storage [28]. In addition, Di (2-ethylhexyl) phthalate (DEHP), a plasticizer found in IV bags, and its metabolites were also elevated [28]. In a recent report, the urinary metabolic profiles of 12 individuals were investigated using ultra high-performance liquid chromatography-tandem mass spectroscopy (UPLC-MS). In the double-blinded, placebo-controlled study, half the participants donated two whole blood units, while half received a sham phlebotomy. After 4 weeks, ABT participants received a transfusion of their RBC concentrates, while the placebo group received a small volume of saline. Five DEHP metabolites were detected for up to three days following ABT [29]. However, DEHP exposure from dietary and occupational sources could not be ruled out [30,31].

Building on these findings, this study used the model of UPLC-MS, certified by Metabolon Inc, to examine both serum and urinary metabolites in healthy individuals who donated and then were reinfused with RBC concentrates of their respective blood. Participants were sampled for blood and urine at three phases of the procedures: prior to donation (baseline), post-donation and up to 7 days post-transfusion. This study investigated the hypothesis that both blood donation and transfusion cause disruption to metabolism, and specific serum and urinary metabolites may act as long-term biomarkers for these procedures. The above hypothesis was investigated at various time points up to 7 days post-transfusion.

## 2. Results

### 2.1. Biomarker Discovery

In total, 66 urine and 66 serum samples were analyzed. 1321 metabolites were found in urine, 912 of which the structural identities were known, while 409 were unknown compounds. A total of 990 metabolites were found in serum, of which 781 were of known identities and 209 compounds were of unknown structural identities.

For initial analysis, metabolites discovered in urine or serum at baseline, post-donation and post-transfusion time-points were pooled together for all participants. The fold changes for all metabolites, comparing baseline vs. post-donation and baseline vs. post-transfusion were calculated and *p* values were produced using a two-tailed *t*-test. Volcano plots were produced by plotting the fold changes against the *p* values (See Figure 1).

In a related analysis, serum and urine metabolites discovered at the final post-donation (D29–D32) and post-transfusion (+168 h) time-points were compared against each other and against the baseline. Volcano plots for these are shown in Figure 2.

### 2.2. Plasticizers

The presence of phthalates and their metabolites have been reported in urine following ABT [27]. In the present study, two phthalate metabolites were identified in urine, but not in serum (Table 1).

The concentrations of both plasticizers showed sharp increases immediately post-donation and post-transfusion (Figure 3). After blood donation, concentrations returned to baseline by D29–32. After blood transfusion, concentrations returned to baseline after 96 h.

### 2.3. PCA Analysis

We performed a PCA for both serum and urine samples separately. The scree plots of the PCA on the serum samples, show that PC 1 and PC 2 explained 16.4% and 13.7% of the variance, respectively. The scree plots of the PCA on the urine samples showed that PC 1 and PC 2 explained 48.1% and 7.4% of the variance, respectively.

The PCA plots showed there was no clear delineation between post-transfusion, post-donation or baseline samples (Figure 4 and Figure 5).

### 2.4. Panel Identification

Using the volcano plot data, metabolites that had a *p* < 0.05 and met a fold change >2 were selected for the panel. These metabolites were significantly different from baseline compared to post-donation and/or post-transfusion in serum and urine. The concentrations of these metabolites were then plotted for all participants across all time-points using Rstudio version 1.4.1103. Unknown compounds were excluded.

#### 2.4.1. Metabolites Altered during the Experiment

A total of 17 known metabolites were significantly altered during the experiment. Of these eight were not sustained in the post-transfusion stage. Four of these were serum metabolites (glycocholate, glycochenodeoxycholate, 12-HETE and lactosyl-N-palmitol-spingosine) and four were urinary (cystathionine, glucuronide of C_10_H_22_O_2_ (10*), glucuronide of C_12_H_22_O_3_ (1*) and enterolactone) (Figure 6). Interestingly, upregulation of serum 12-HETE was a particularly robust marker for blood donation, which returned to baseline levels within 96 h post-transfusion.

#### 2.4.2. Selected Metabolite Panel

The remaining nine metabolites (seven serum and two urinary) were significantly altered up to Day 7 (+168 h) post-transfusion. Of these, three serum metabolites (glycoursodeoxycholate, downregulated; guanosine and inosine, upregulated) were significantly altered both post-donation and post-transfusion, compared to baseline (Figure 7I–III). The other four serum metabolites in this panel (S-allcysteine, 17-alphahydroxypregnenalone 3, Glutamine conjugate of C_6_H_10_O_2_ (2)* and Sphinganine) were only altered post-transfusion (Figure 7IV–VII).

In urine only erythronate remained upregulated in both the post-donation and post-transfusion phases, compared to baseline. Urinary isocitrate, on the other hand, gradually increased post-transfusion, achieving significance 7 days post the procedure (Figure 8).

After *p* value adjustments using the Benjamini-Yekutieli method to account for multiple testing comparisons, serum inosine remained significant. However, since this is a proof-of-concept study the panel of metabolites identified (Figure 7 and Figure 8) are reported for future studies and validation.

## 3. Discussion

The discovery of robust markers able to reliably and reproducibly detect ABT has been particularly challenging. Recent advances in the omics, especially metabolomics, offers new avenues of investigation. Using an advanced model of untargeted metabolomics, significant changes were noted in a panel of nine metabolites (seven in serum, two in urine), which together make up a reliable panel of makers of autologous blood transfusion up to 7 days post-transfusion in healthy males. Notably, serum guanosine, inosine and sphinganine and urine isocitrate and erythronate were upregulated, while serum glycourdeoxycholate, S-allylcysteine, 17-alphahydroxypregnenalone 3 and Glutamine conjugate of C_6_H_10_O_2_ (2)* were downregulated. The seventh day post-transfusion time point allowed for the analysis of changes, which were independent of storage duration, concentration or dilution effects.

The selected panel did not include plasticizers and components of storage preservatives. Two plasticizers were transiently elevated in urine post-donation and post-transfusion, but levels had returned to baseline by the seventh day. Elevation in urinary plasticizers have been reported previously [29] and they can be present in plastics used for urine collection [32]. There were no changes in plasticizers in the serum samples in this study, although they have been seen in plasma immediately post-transfusion previously [28]. These differences may be related to the type of plastics used for RBC storage and how plasticized they are with different compounds [33]. These differences highlight the dependence of the presence of plasticizers on storage conditions, which can be variable and modifiable, and therefore unreliable as markers of autologous transfusion.

Unlike the previous studies, two additional urinary metabolites, which are not plasticizers were identified here. The differences with the previous study may be related to the models of metabolomic analysis adopted. The previous metabolomic study which was also untargeted [29] utilized dilute and shoot procedure followed by chromatographic separation through C-18 column in UHPLC and analysis was by electrospray ionization (ESI) in positive and negative modes. The study involved an ion centric approach for the analysis of data. In the ion centric approach, the investigators took all the features created by the mass spectrometry (a single compound can produce multiple features), and then applied some statistical filtering and compared the two groups. From the comparison they obtained a list of features which differentiated the groups. From this they elucidated the structure by re-running samples to obtain fragmentation data which was compared against public databases and then by purchasing candidate molecules and running these to compare against the experimental fragmentation data. The ion centric approach worked reasonably well [29] but this is potentially because it was a well-controlled study with matched participants. It is difficult to predict whether this approach will work for a study with more “free range” human samples.

In contrast, the present study involved chromatographic separation through four different injections as explained in the method, through two different columns (C-18 column and HILIC column). The use of additional amide column helps to acquire a wider range of metabolites, which would not be possible through the C-18 column alone. Secondly, the present study utilized a chemocentric approach which identified some or all of the doping markers immediately, since the fragmentation data and the library built on 3300 authentic standards already exists. Therefore, the re-analysis step to obtain fragmentation data to compare to databases and the acquisition or synthesis of candidate molecules was not required. These differences in approach may at least in part explain the discovery of additional metabolites in the present study. Our model showed good concordance in the data generated by reanalysis of samples in both our laboratory (HD4 ADLQ) and Metabolon Inc (HD4 Durham). Both ADLQ and the Durham HD4 platforms showed good recovery of metabolites from samples stored for 6–7 years. Thus, our model was stringently tested for both accuracy and reproducibility, which is fundamental for an acceptable anti-doping biomarker discovery platform. The slightly lower correlation between HD4 ADLQ and HD2 compared with HD4 Durham and HD2 may be explained by a difference in the matrix (EDTA versus heparin plasma).

The RBC concentrates were stored in the conventional saline-adenine-glucose-mannitol (SAGM) preservative, and neither the components of this preservative nor their metabolites were seen nor included in this panel. Interestingly, the components of SAGM do not normally impact the metabolic shift associated with RBC storage [34] consistent with present data. Although RBC storage can increase purine metabolites upon transfusion [35] such effect is short-lived and would not explain the elevation in serum inosine levels observed on the seventh day post-transfusion. Moreover, the level of guanosine, another nucleotide intermediate, which is not known to change with storage in SAGM [36], was also elevated on the seventh day post-transfusion, reinforcing the suggestion of an effect which is independent of storage.

The members of this novel biomarker panel come from various metabolic pathways. Glycoursodeoxycholate is a bile acid glycine conjugate derived from ursoodeoxycholic acid, which is involved in digestion of carbohydrates and dietary lipids. Guanosine and inosine are important for transport of nucleosides and free purine and pyrimidine bases across the plasma membrane. Guanosine is comprised of guanine attached to a ribose (ribofuranose) ring via a β-N9-glycosidic bond [37], while inosine is generated intracellularly and extracellularly by deamination of adenosine or intracellularly through the action of 5′-nucleotidase on inosine monophosphate [38]. Erythronate and isocitrate are important for pyruvate metabolism and the citric acid cycle. Erythronate is derived from erythronic acid which is formed when N-acetyl-D-glucosamine (GlcNAc) is oxidized. GlcNAc is a constituent of hyaluronic acid (HA), a polysaccharide, and may be derived from glycated proteins or from degradation of ascorbic acid. Isocitrate is an intermediate metabolite generated in the TCA cycle by the actions of the NAD(+)- or NADP(+)-dependent enzymes, isocitrate dehydrogenases, which convert it to alpha-ketoglutarate by oxidation and decarboxylation reactions [39]. Sphinganine, also termed dihydrosphingosine, is biosynthesized by a decarboxylating condensation of serine with palmitoyl-CoA to form a keto intermediate, which is then reduced by NADPH [40]. S-Allylcysteine is an organosulfur derivative of the amino acid cysteine, while 17α-Hydroxypregnenolone is a pregnane (C21) steroid that is obtained by hydroxylation of pregnenolone at the C17α position by a mitochondrial cytochrome P450 enzyme, 17α-hydroxylase (CYP17A1) [41]. Considered a prohormone in the formation of dehydroepiandrosterone (DHEA), 17α-Hydroxypregnenolone is itself a prohormone of the sex steroids. Glutamine is a nonessential amino acid, which can donate the ammonia on its side chain to the formation of urea and to purines (necessary for the synthesis of nucleic acids). The conversion of glutamic acid to glutamate by glutamine synthase is of central importance in the regulation of toxic levels of ammonia in the body. It is therefore noteworthy that the panel of nine metabolites identified in this study can be associated with metabolic activities and not with storage, concentration or dilution effect of the procedure. Together this panel therefore represents a reliable marker of ABT.

One of the limitations of this study was the small sample size (*n* = 6). A greater number of subjects could provide more information to cluster the data into the three grouped time points. Furthermore, additional subjects would allow for a supervised discriminant analysis, such as PLS-DA. A greater number of subjects would also allow for testing accuracy of a model, using PLS-DA, by using a training and testing data set. These limitations were somewhat mitigated by the longitudinal sampling of each subject who also served as their own control over the period of the experiment. The use of only male volunteers may also limit the wider application of the findings. In addition, prolonging the sampling times post-transfusion would be necessary to determine the utility of these markers over a longer period.

In conclusion, this is the first proof of concept study that has identified a panel of markers that stayed significantly altered for up to 7 days after autologous blood transfusion. The metabolites identified here will need to be validated in a separate, future study. This offers new avenues for the future in biomarker discovery for ABT.

## 4. Materials and Methods

### 4.1. Ethical Approval

The study was approved by the Ministry of Public Health registered Institutional Review Board of the Hamad Medical Corporation (Protocol No.: MRC-02-18-070).

### 4.2. Study Participants

Six healthy, male volunteers participated in this study. The mean ± standard deviation for age, height and weight were 37 ± 6 years, 180 ± 6 cm and 84 ± 7 kg, respectively. All volunteers gave written consent and were subjected to an initial health check before participation. Participants also completed an online questionnaire as per the National Blood Donor Centre requirements, to confirm they fulfilled all inclusion criteria. Five participants practiced sports 3–5 times per week, while one was not physically active. Three participants had an A+ blood group, two were B+ and one was O+.

### 4.3. Study Design

Participants were assessed, and blood and urine samples were collected prior to blood donation (D0) as a “baseline” for each individual. Immediately after, blood was withdrawn using an indwelling cannula. An amount of 1 unit (450 mL) was stored for autologous blood transfusion (ABT). Further blood and urine samples were collected at time points (D2-D32 post-donation and 1–168 h post-transfusion) shown in the schema in Figure 9. The donated samples were stored for 35–36 days prior to being transfused, and the sampling was carried out for up to 7 days (+168 h) post-transfusion.

### 4.4. Blood and Urine Sampling

Up to 40 mL of blood was collected via an antecubital vein puncture using a 21-gauge needle. Tubes were stored in a cool-box at 8–14 °C as monitored by a thermometer. The tubes were centrifuged within one hour of collection at 1850 g for 20 min at 8° (Universal 320R centrifuge C (Hettich, Germany). Midstream urine samples were also collected at the sample time-points.

### 4.5. Blood Donation and Storage procedure

Standard blood donation protocols of the hospitals were adopted [21]. An amount of 1 unit of blood was collected in a 600 mL bag containing 63 mL citrate phosphate dextrose (CPD) solution (Terumo, Millbrok, UK) via an antecubital vein puncture using a 16-gauge needle. The blood was then leucocyte reduced, and RBCs were separated from plasma and buffy coat, UV irradiated to eliminate potential pathogens and stored in 100 mL saline, adenine, glucose, mannitol (SAGM) preservative.

### 4.6. Blood Transfusion

The transfusion of the RBC concentrate was performed either after 35 or 36 days of storage depending on the availability of the facilities. Hospital standard procedures, which included cross-matching of the blood groups for safety purposes were followed. The transfusion was then performed via a 22 G catheter at a flowrate of 150 mL/h over approximately two hours.

### 4.7. Sample Analysis

Each participant donated 11 serum samples and 11 urine samples. A total of 132 samples were received for metabolomic analysis. Sample preparation, processing and QC generation were completed at ADLQ. The final QC review and curation were completed by Metabolon, USA.

Samples were prepared using the automated MicroLab STAR^®^ system (Hamilton Bonaduz AG, Bonaduz, Switzerland). Several recovery standards were added prior to the first step in the extraction process for QC purposes. Proteins were precipitated with methanol under vigorous shaking for 2 min (Glen Mills GenoGrinder 2000, Glen Mills Inc, Clifton, NJ, USA) followed by centrifugation. The resulting extract was divided into five fractions; the sample extracts were stored overnight under nitrogen before preparation for analysis.

### 4.8. Validation of Metabolomics Method

The methodology used in this study has been validated for a variety of analytical parameters at Metabolon, USA for global biochemical profiling platform. The parameters validated were intra and inter precision, carryover, method comparison (accuracy), limit of detection, linearity, matrix effect, recovery, reproducibility and sample stability testing [42,43], and the same analytical methods are applied in the present study. The specific instruments used in the present study were qualified before use for a variety of parameters including limit of detection, precision and biological variation [42,43].

The platform used for the collection of this data was further qualified for stability of response (+/− 15%) in internal standards over multiple days. Equivalency to existing instrumentation in Metabolon, USA was demonstrated in biologically diverse samples of serum and urine. The equivalency was demonstrated via both absolute number of identified compounds (+/− 10%) and reproducibility within technical replicates (median RSD for all endogenous compounds of <10%) post curation of the data.

Finally, a set of >300 samples from a previous study (QMDiab) which had been analyzed in North Carolina (Metabolon, NC, USA) were run on the instruments in ADLQ and <770 common compounds were identified with an average R value of 0.70 and 208 compounds with an R ≥ 0.90 (data in press).

### 4.9. Ultrahigh Performance Liquid Chromatography-Tandem Mass Spectroscopy (UPLC-MS/MS)

The study utilized Waters ACQUITY UPLC and a Thermo Scientific Q-Exactive high resolution/accurate mass spectrometer (Thermo Scientific, Bremen, Germany) interfaced with a heated electrospray ionization (HESI-II) source and Orbitrap mass analyzer operated at 35,000 mass resolution (Thermo Scientific, Bremen, Germany) as previously described [44]. The extracted samples were reconstituted in solvents compatible to each of the four methods. Each reconstitution solvent contained a series of standards at fixed concentrations to ensure injection and chromatographic consistency. One aliquot was analyzed using acidic positive ion conditions, chromatographically optimized for more hydrophilic compounds. The extract was gradient eluted from a C18 column (Waters UPLC BEH C18–2.1 mm × 100 mm, 1.7 µm) using water and methanol, containing 0.05% perfluoropentanoic acid (PFPA) and 0.1% formic acid (FA). A second aliquot was analyzed using acidic positive ion conditions, and chromatographically optimized for more hydrophobic compounds. The extract was gradient eluted from the C18 column using methanol, acetonitrile, water, 0.05% PFPA and 0.01% FA and was operated at an overall higher organic content. A third aliquot was analyzed using basic negative ion optimized conditions using a separate dedicated C18 column. The basic extracts were gradient eluted from the column using methanol and water, with 6.5 mM Ammonium Bicarbonate at pH 8. The fourth aliquot was analyzed via negative ionization following elution from a HILIC column (Waters UPLC BEH Amide 2.1 mm × 150 mm, 1.7 µm) using a gradient consisting of water and acetonitrile with 10 mM Ammonium Formate, pH 10.8. The MS analysis alternated between MS and data-dependent MSn scans using dynamic exclusion. The scan range varied slighted between methods but covered 70–1000 m/z.

### 4.10. Data Extraction and Compound Identification

The informatics system consisted of four major components, the laboratory information management system (LIMS), the data extraction and peak-identification software, data processing tools for QC and compound identification, and a collection of information interpretation and visualization tools for use by data analysts. The hardware and software foundations for these informatics components were the LAN backbone, and a database server running Oracle 10.2.0.1 Enterprise Edition.

Raw data were extracted, peak identified and QC processed using Metabolon’s hardware and software [45]. Compounds were identified by comparison to library entries of purified standards or recurrent unknown entities. The library matching was based on the authenticated standards that contains the retention time/index (RI), mass to charge ratio (m/z), and chromatographic data (including MS/MS spectral data) on all molecules present in the library. The biochemical identifications were based on three criteria: retention index within a narrow RI window of the proposed identification, accurate mass match to the library +/− 10 ppm, and the MS/MS forward and reverse scores between the experimental data and authentic standards.

The QC and curation processes were designed to ensure accurate and consistent identification of true chemical entities, and to remove those representing system artifacts, mis-assignments, and background noise. Library matches for each compound were checked for each sample and corrected if necessary.

Peaks were quantified using area-under-the-curve. A data normalization step was performed to correct variations resulting from instrument inter-day tuning differences.

### 4.11. Statistics and Bioinformatics

The selection of the list of metabolites was done using both Volcano plots and PCA. Since not all metabolites produced a reading for every subject at every time point, we initially compared pooled baseline data to the pooled data post-donation and post-transfusion to generate a list of metabolites most altered by the procedures while keeping the results comparable between metabolites. Changes at the individual time points were then compared using paired t-tests. The volcano plots show raw *p*-values.

During this initial process, adjusted *p* values were not used as the selected metabolites were not subject to outliers. For further stringency the Benjamini-Yekutieli method was used to account for multiple testing comparisons.

A supervised PCA was not possible in this case, as in order to test the viability of a PCA we would require both a training and test set. Typically, 30% of the data would need to be retained as a test set, this would be four subjects for the training set and two for the test set. It is not possible to have a robust inference on a test set of the remaining two subjects.

#### 4.11.1. Volcano Plots

For each metabolite we compared the samples taken at two different time-points. For chemical i, the samples taken at one time point was called $X_{1}^{\left(i\right)}$, and the samples taken at another time point was called $X_{2}^{\left(i\right)}.$ We calculated the *p*-value, $p^{\left(i\right)},$ of the two-tailed t-test using $X_{1}^{\left(i\right)}$ and $X_{2}^{\left(i\right)}.$ The *y*-axis of the volcano plots was the $-Log_{10}\left(p^{\left(i\right)}\right).$ The *x*-axis of the volcano plot was $Log_{2}\left(\bar{X_{2}^{\left(i\right)}}/\bar{X_{1}^{\left(i\right)}}\right)$, where $\bar{X_{1}^{\left(i\right)}}$ and $\bar{X_{2}^{\left(i\right)}}$ are the means of $X_{1}^{\left(i\right)}$ and $X_{2}^{\left(i\right)}$, respectively. In the plots, points closer to the origin are lighter and get darker as points get further from the origin.

#### 4.11.2. Principal Component Analysis

The effects of ABT on metabolites was determined by principal component analysis (PCA). Each principal component (PC) is a linear combination of metabolite values, where the $i^{th}$ PC explains a lower amount of variance than the previous $i^{th}-1$ PCs. The data matrix X was mean centred and unit variance scaled. Each of the subjects gave one baseline sample, four post-donation samples, and six post-transfusion samples. The eigenvectors of $X^{T}X$ provide the PCs loadings, where the eigenvalues show the variance explained in the data. An assumption we have made is that the variance in the data is mostly explained by the samples taken at different time points and not between subjects.

## Figures and Tables

**Figure 1 metabolites-12-00425-f001:**
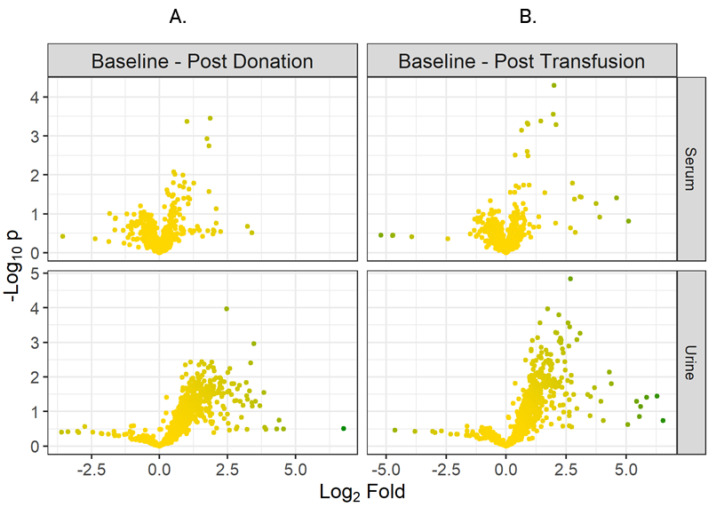
Volcano Plots (**A**) comparing Baseline vs. Post-donation in serum and urine, (**B**) comparing Baseline vs. Post-transfusion in serum and urine.

**Figure 2 metabolites-12-00425-f002:**
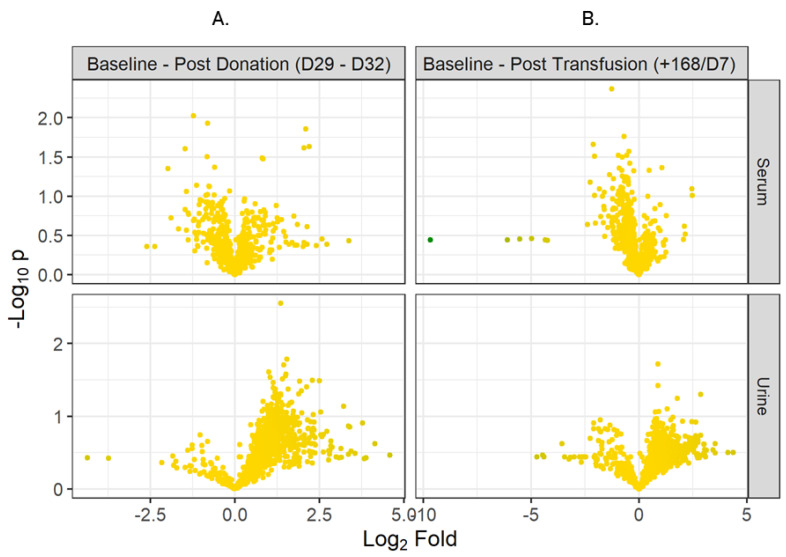
Volcano Plots (**A**) comparing Baseline vs. Post-donation (D29–32) in serum and urine, (**B**) comparing Baseline vs. Post-transfusion (+168 h) in serum and urine.

**Figure 3 metabolites-12-00425-f003:**
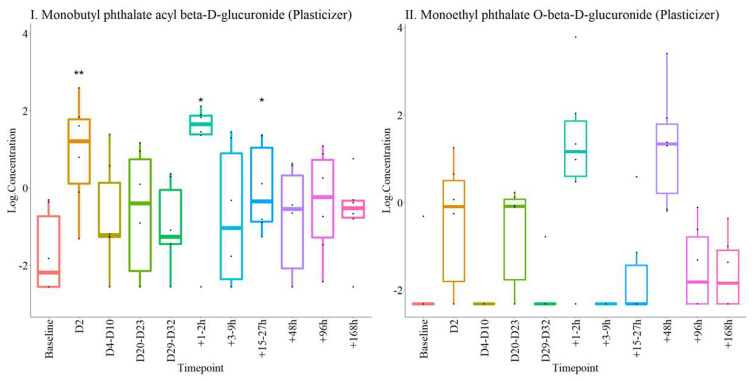
Box-plots detailing concentrations of plasticizers at all timepoints. I. Monobutyl phthalate acyl beta-D-glucuronide and II. Monoethyl phthalate O-beta-D-glucuronide. * *p* < 0.05, ** *p* < 0.01 compared with baseline.

**Figure 4 metabolites-12-00425-f004:**
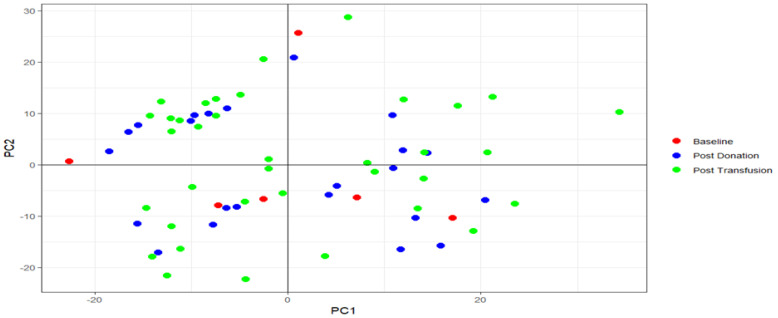
PCA plot of serum samples. Plot of PC 1 vs. PC 2.

**Figure 5 metabolites-12-00425-f005:**
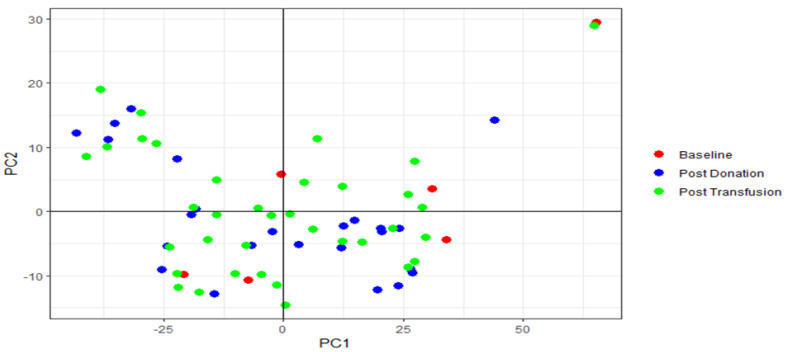
PCA plot of urine samples. Plot of PC 1 vs. PC 2.

**Figure 6 metabolites-12-00425-f006:**
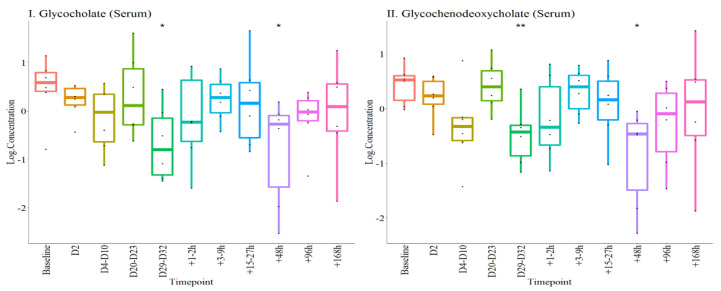
Concentrations of eight metabolites (four in serum and four in urine) altered at different time points before day 7 post-transfusion. The log concentrations at different time-points are plotted for each metabolite as labelled in I to VII1. * *p* < 0.05, ** *p* < 0.01, compared with baseline.

**Figure 7 metabolites-12-00425-f007:**
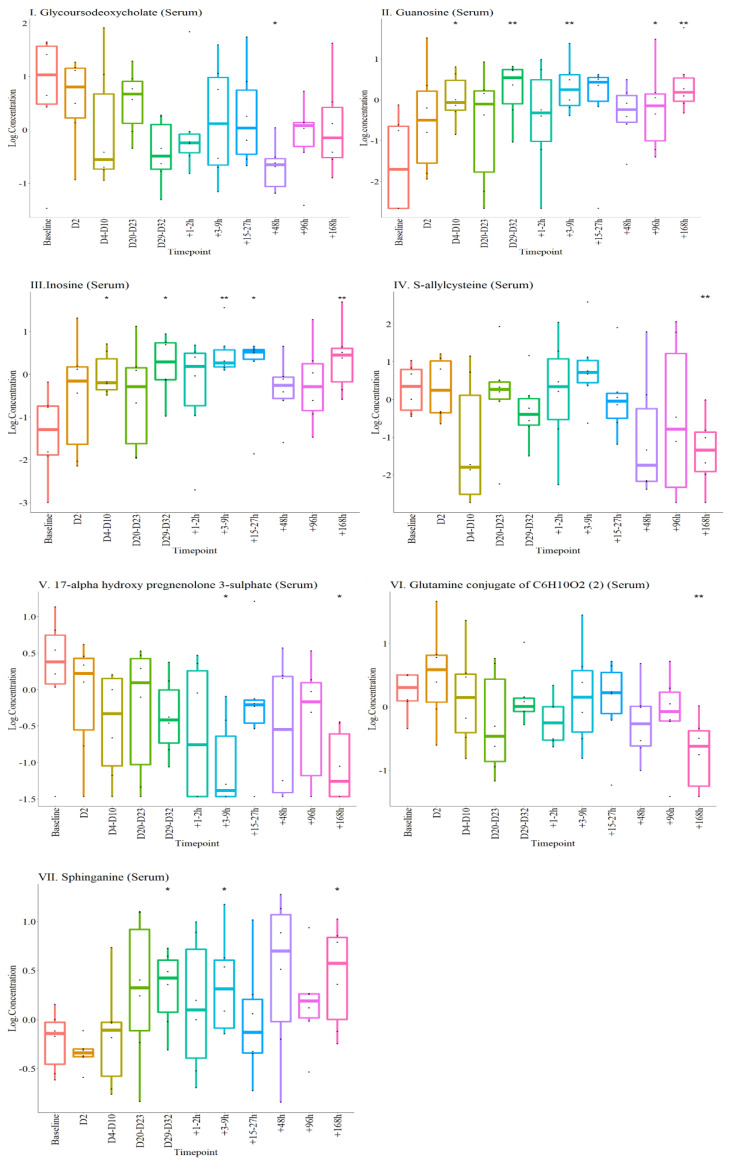
Serum concentrations of 7 metabolites altered at different time points during the experiments. The log concentrations at different time-points are plotted for each metabolite as labelled in I to VII. * *p* < 0.05, ** *p* < 0.01, compared with baseline.

**Figure 8 metabolites-12-00425-f008:**
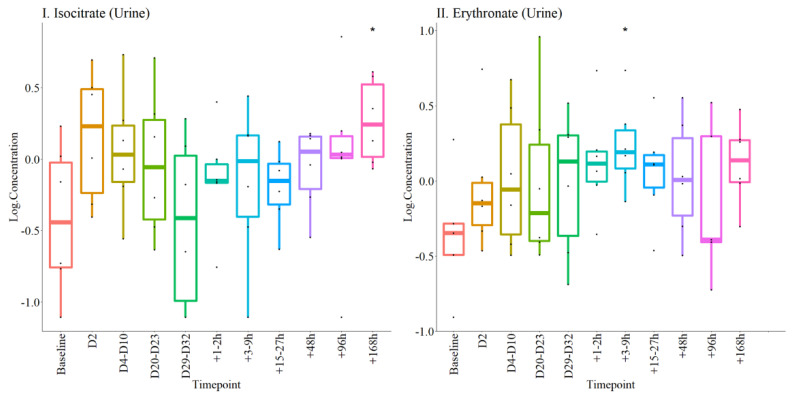
Urine concentrations of metabolites altered at different time points during the experiments. The log concentrations of each metabolite at different time-points are plotted separately as labelled in I and II. * *p* < 0.05, compared with baseline.

**Figure 9 metabolites-12-00425-f009:**
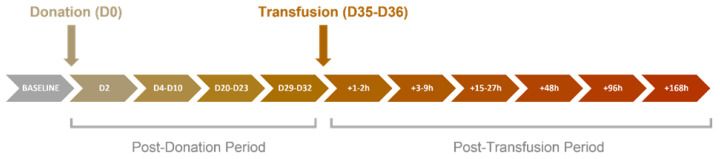
Schema of study design and sample collection (h = hours, D = day).

**Table 1 metabolites-12-00425-t001:** Phthalates identified in urine in post-donation and post-Transfusion phases.

Baseline—Post Donation (D29–D32)	*p*	Fold
Monoethyl phthalate O-beta-D-glucuronide	0.31	4.12
Monobutyl phthalate acyl-beta-D-glucuronide	0.10	5.01
**Baseline—Post Transfusion (+168 h/D7)**		
Monoethyl phthalate O-beta-D-glucuronide	0.35	3.19
Monobutyl phthalate acyl-beta-D-glucuronide	0.31	5.84

## Data Availability

Data is contained within the article.

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
