# Peer review of "Untargeted Metabolomics Identifies a Novel Panel of Markers for Autologous Blood Transfusion"

_metabolites, 2022, doi:10.3390/metabo12050425_

Round 1
Reviewer 1 Report
The manuscript entitled "Untargeted metabolomics identifies a novel panel of markers of autologus blood transfusion" describes a study on urine and serum putative markers of ABT procedure. The manuscript is well planned and written in a clear manner, however some minor corrections in English are required. I have only minor suggestions and questions. I can highly recommend the manuscript for publication after some corrections.
- The sample collection time points should be more clearly specified in the text.
- Can the phtalates observed in urine be present in containers used for urine collection?
- Were the statistical methods selected with the assumption that samples were collected from one subject in different time points?
Reviewer 2 Report
The authors present a novel panel of serum and urine biomarkers of autologous blood transfusion. The relevance of the study is well described in the introduction and the methodology is sound for this initial pilot study. The limitations of the study are well described. I have mostly minor comments and one major concern:
Major: For the panel identification (section 2.5 and Figures 5-7) were the p-values adjusted for multiple comparisons? Please justify your decision, regardless of if the p-values were adjusted or not.
Minor comments:
In line 88, the authors state that "this study used a validated model of UPLC-MS". Please provide a reference for this model.
I am not sure what section 2.1 on the validation of the metabolomics method adds to the manuscript. The justification does not occur until the discussion (lines 244-246) and so it is a little confusing to read it immediately after the Introduction. It may be more suited to the Methods section.
In lines 98-99, the authors refer to the QMDiab study. Please provide a reference or references for this study.
Section 2.4.2 appears to be mislabeled; I assume it should be section 2.5.2. Please fix this.
The word 'aliquots' is misspelled.
Reviewer 3 Report
In this work, the authors applied untargeted metabolic profiling of serum and urine samples to identify biomarkers for autologous blood transfusion.
Major comments:
- The sample size is too short to extract useful (generalizable) conclusions and increased the margin of false positive detection of biomarkers.
- The authors describe the analytical method as validated (see e.g., Abstract or line 241). However, details describing the validation process and criteria are insufficient. Whereas results described in section 2-1 support the reproducibility of the results and the comparability among the metabolic profiles of EDTA and heparinized samples, data reported did not assess the accuracy or the precision of the method among other analytical parameters that should be assessed during the validation of a method. Furthermore, the reported reproducibility test was not carried out for urine or serum samples. In line 244, the authors describe that ‘Thus, our model was stringently validated for both accuracy and reproducibility over several years, which is fundamental for an acceptable anti-doping testing model.’ I disagree with that description. The accuracy of the method was not assessed and so, the method cannot be considered as validated. While I acknowledge that validating an unsupervised analytical method aimed at the discovery of novel biomarkers might be a very hard task (if feasible), the results reported do not follow the World Anti-Doping Code International Standard for Laboratories (ISL) developed as part of the WADA World Anti-Doping Program.
- Further information should be provided to support that the overlap between HD2 and HD4 is ‘comparable in large part’ (l.106). PCA could be used, for example, to identify main sources of variation between experiments, and further information regarding the matched and not-matched variables (e.g., class, subclass, intensity) would be useful.
- Why did the authors use an unsupervised multivariate analysis (PCA) for the identification of biomarkers? In section 2.2, the fold changes for all metabolites were compared as baseline vs post-donation and baseline vs post-transfusion using a two-tailed t-test (without correction for multiple testing), and the same approach is later used in section 2.5 Panel identification.
- Besides reference 43, a complete description of the analytical methods (e.g., instruments, columns, chromatographic and instrument parameters, characteristics of the spectral library used for annotation, the algorithm used for peak table generation, data clean up, QA/QC) should be provided to ensure reproducibility.
Minor comments:
- Although widely used, in my opinion, no information can be extracted from the volcano plots shown in Figure 1. It would be enough to describe the number of features above/below thresholds in the discussion.
- No correction for multiple testing was applied for biomarker selection.
- Did the authors normalize the urine concentrations?
- Please increase the ‘dot’ size in Figure 4. I would also recommend including confidence intervals in the PCA scores plot.
